# Sleep Benefits Prose Memory Consolidation in University Students

**DOI:** 10.3390/brainsci15030265

**Published:** 2025-03-01

**Authors:** Francesca Conte, Serena Malloggi, Oreste De Rosa, Gianluca Ficca, Stefania Righi, Maria Pia Viggiano, Fiorenza Giganti

**Affiliations:** 1Department of Psychology, University of Campania L. Vanvitelli, Viale Ellittico 31, 81100 Caserta, Italy; francesca.conte@unicampania.it (F.C.); oreste.derosa@unicampania.it (O.D.R.); gianluca.ficca@unicampania.it (G.F.); 2Department of Neurofarba, University of Florence, Via di San Salvi 12, 50135 Florence, Italy; stefania.righi@unifi.it (S.R.); mariapia.viggiano@unifi.it (M.P.V.); fiorenza.giganti@unifi.it (F.G.)

**Keywords:** prose memory, memory consolidation, sleep effect, sleep

## Abstract

**Background/Objectives**: Sleep plays a pivotal role in memory consolidation, especially for declarative memory. While extensive research has examined sleep’s impact on simple declarative materials, such as word lists, its effect on more complex narrative passages remains less studied. This study aimed to investigate the effect of sleep on prose memory consolidation. **Methods**: In a within-subjects design, 10 healthy university students (6 F, 4 M; age range = 19–29; generation ratio = 0.8) learned prose passages and were tested both immediately and after a retention interval spent either asleep or awake. **Results**: The analyses revealed a positive effect of sleep, with the participants recalling more textual details in the Sleep condition (R^2^ conditional = 0.269). Correlational analyses further supported this outcome, showing a positive association between specific sleep features (i.e., sleep cycles) and prose memory performance (r^2^ = 0.56). **Conclusions**: Our finding suggests that sleep facilitates the consolidation of complex declarative memory traces, counteracting the decay that occurs during wakefulness; correlational analyses further support this outcome. Overall, these findings underscore the importance of sleep for everyday learning and comprehension, particularly when processing complex textual information.

## 1. Introduction

There is a broad consensus on the existence of a relationship between sleep and memory (for reviews, see [1,2]). Numerous studies have shown that sleep supports memory trace consolidation more than wakefulness, a phenomenon known in the literature as the ‘sleep effect’ [3]. The sleep effect has been demonstrated for various types of information, like procedural and declarative memories (for reviews, [1,4]). Regarding procedural memory, studies have shown that REM sleep is involved in the consolidation and transfer of novel implicit knowledge [5,6], a process that seems to be enhanced when subjects use metacognitive strategies [7]. In parallel, various studies have shown that NREM sleep promotes the consolidation and the reorganization of declarative memory traces [4,8]. Specifically, slow oscillations and slow-wave sleep (SWS) [9,10,11], as well as NREM sleep features such as sleep spindles, are believed to be particularly beneficial for hippocampus-dependent declarative memory consolidation [12].

Declarative memory encompasses both memories regarding personal experiences (episodic memory) and memories regarding general knowledge (semantic memory). In sleep studies on declarative memory, participants are typically asked to memorize stimuli, such as word lists or paired associates, before night sleep and to retrieve them the following morning [13].

While there is extensive research on the impact of sleep on simple declarative memory tasks, its effect on the consolidation of more complex memory traces, such as narrative passages, has been relatively neglected. Prose memory is a type of long-term declarative memory referring to the ability to remember the content of text materials with different lengths or complexity. We often use prose memory in daily contexts, such as educational ones. The first studies on the topic are outdated and mainly focused on investigating the contribution of REM sleep to the consolidation of brief prose passages, such as a 162-word prose passage created ad hoc [14,15], and the War of the Ghosts (from [16]). Through selective REM sleep deprivation paradigms, the authors showed that REM sleep is implicated in prose memory consolidation. However, the sleep effect for prose memory was not evaluated, and no wake–control conditions were included. More recently, other studies on student populations focused on the impact of prolonged sleep loss on the acquisition [17] and retrieval [18], but not the consolidation, of prose materials. Furthermore, we have recently shown that the pre-sleep learning of theatrical monologues improves subsequent sleep quality measures [19]. While changes in post-learning sleep parameters are generally ascribed to the involvement of those parameters in memory consolidation, these data represent only indirect proof of a sleep-related processing of this kind of memory.

Only Bäuml and colleagues [20] have specifically addressed the effect of sleep on prose memory consolidation. Using a between-subjects design, the authors found better recall of texts following a night of sleep than after a waking interval. However, the study did not account for potential differences in encoding between the sleep and the wake conditions, a variable that could impact later performance [21]. Therefore, it cannot be ascertained whether the enhanced performance observed after sleep was due to differences between the conditions in encoding rather than to a more effective consolidation process. Furthermore, the absence of baseline performance measures does not allow to evaluate the degree of memory deterioration over the retention intervals (for the importance of evaluating differences in encoding performance between sleep and wake conditions, see [4,22].

Here, we aimed to expand this scarce literature on sleep and prose memory: in a within-subjects design, we compared memory performance for prose texts after sleep and wake. Performance was assessed both immediately (just after the acquisition phase) and after the retention interval spent asleep or awake in order to (a) exclude that possible between-conditions differences at delayed recall could depend on differences in acquisition levels rather than on the efficacy of the consolidation process; (b) evaluate the degree of memory deterioration across the sleep and wake retention periods. In line with the existing literature on sleep-related memory consolidation, the present study explored the impact of sleep on the retention of prose material. We hypothesized that sleep would facilitate the consolidation of these memories, leading to improved recall performance compared to wakefulness.

## 2. Materials and Methods

### 2.1. Participants

The study was proposed to the university student population through an online advertisement on major social media platforms (WhatsApp and Instagram). Potential participants were previously asked to fill out a web-based set of questionnaires through the Google Form platform, i.e., the Pittsburgh Sleep Quality Index (PSQI; [23]), the Beck Depression Inventory-II (BDI-II; [24]), and the Beck Anxiety Inventory (BAI; [24]), and ad hoc questions assessing demographic data, general medical conditions, and health habits.

The inclusion criteria were status of university student, absence of any relevant somatic or psychiatric disorder, absence of clinically significant depression and anxiety symptoms (BDI-II score ≤ 29; BAI score ≤ 25), no history of drug or alcohol abuse, good sleep quality (PSQI < 5), having a regular sleep–wake pattern (e.g., individuals with irregular study or working habits such as shift work were excluded), no use of psychoactive medication or alcohol at bedtime.

We ran an a priori power analysis (through G*Power, version 3.1) with 1 group, 2 measurements, alpha = 0.05, within-factors rmANOVA, which indicated that a sample of 34 subjects was necessary to obtain an effect size of 0.25 with 80% power. During the time allotted for recruitment, 40 volunteers expressed their interest in the study, while 35 of them finally gave their consent to participate. Of them, 24 potential participants had to be excluded because of the presence of sleep disorders or poor sleep quality (PSQI score > 5), the presence of anxiety and/or depressive symptoms (BAI score > 25, BDI-II score > 29), temporary inability to reach the Sleep Lab. In addition, we had to exclude one participant because he did not adhere to his habitual sleep schedule during the study. Therefore, the final sample comprised 10 university students (6 F, 4 M; mean age = 23.5 ± 3.18 years; age range = 19–29; generation ratio = 0.8).

An a posteriori power analysis (within-factors rmANOVA, with 1 group, 2 measurements, alpha = 0.05, effect size f of 0.45, total sample size of 10 participants) revealed a study power of 33%.

The study design was submitted to the Ethical Committee of the University of Florence, which approved the research (code 363/2024; date 11 December 2024) and certified that the involvement of human participants was performed according to acceptable standards.

### 2.2. Procedure

In a within-subjects design, all participants underwent a Sleep and a Wake condition in which a learning phase was followed by a ≈9 h retention interval spent, respectively, asleep or awake and then by a re-test. The order of the conditions was balanced between participants. The experiments were conducted during working days, in separate weeks. Figure 1 displays the study procedure.

The Sleep condition experiment was conducted at the participant’s home. The experimenter arrived there in the evening, about 40 min before the participant’s habitual bedtime, and immediately administered the learning phase of the memory task. Then, the experimenter proceeded to setting up the Dreem Headband (DH) for sleep recording, and the participant was instructed to go to sleep. The re-test phase of the memory task was performed in the morning, 30 min after awakening to allow for sleep inertia dissipation. On the night before the Sleep condition, each participant was administered a night of habituation to the DH.

In the Wake condition, the learning phase was conducted at the Sleep Lab in the morning. The re-test phase was performed in the afternoon, after a retention period corresponding to the participant’s habitual sleep duration. During this interval, the participants were requested to refrain from engaging in cognitively demanding activities.

In both conditions, at the beginning of the learning and of the re-test phases, the participants reported their sleepiness level by filling out the Karolinska Sleepiness Scale questionnaire [25]. Moreover, at the end of the learning phase, the participants were required to answer a set of questions addressing, on a 7-point scale, their level of interest in the text they just studied (‘How interesting was the text you just read?’; response alternatives ranging from 1 = very interesting to 7 = very boring), the ease of reading (‘How easy was it to read?’; response alternatives ranging from 1 = very easy to 7 = very difficult), their previous knowledge of the text topic (‘Did you have any previous knowledge about the topic of the text?’; response alternatives ranging from 1 = very poor to 7 = very high). Moreover, a final question was asked in order to ascertain that the participants had not read the text before (‘Had you ever read this text before?’; response alternatives: yes, no).

### 2.3. Prose Memory Task

The task consisted in studying one of two texts (text 1: ‘The Sun’; text 2: ‘The Sea Otters’, one for each condition), selected from the reading comprehension section of a test-preparation book for the Test of English as a Foreign Language test manual (TOEFL; [26]) and already used in the study of Roediger & Karpicke [27]. The texts were translated into Italian by two experts. The length of the texts was 247 and 308 words, respectively. Since both texts are currently used to assess language skills in young adult high school students, we considered them to be easy to understand and to learn for the university study sample, attending higher level courses of study.

The two texts were assigned to the two conditions in a balanced order between the participants. During the learning phase, the participants were administered in written form one of the two texts and instructed to study it for 5 min. Specifically, the participants were asked not to memorize the prose passages word by word, but to report as many text details as possible. Then, the subjects were asked to complete a 2 min distractor task (i.e., an arithmetic task). Finally, a free recall test (‘immediate test’) was administered, in which the participants had 10 min to write down on a blank piece of paper anything they remembered from the previously learned text. The same free recall test was performed as a re-test (‘delayed test’) after the retention interval spent in sleep or wakefulness.

### 2.4. Sleep Recordings

Sleep was monitored through a DREEM Headband (DH), a wireless device that records, stores, and analyses physiological data in real time. Physiological signals are recorded by means of three types of sensors: (1) brain cortical activity through 5 EEG electrodes yielding seven derivations (FpZ-O1, FpZ-O2, FpZ-F7, F8-F7, F7-O1, F8-O2, FpZ-F8; 250 Hz with a 0.4–35 Hz bandpass filter); (2) movements, position, and breathing frequency through a 3D accelerometer located over the head; (3) heart rate via a red–infrared pulse oximeter located in the frontal band. The DH is an alternative to standard polysomnographic recording and has been considered an adequate device for high-quality large-scale longitudinal sleep studies in the home or laboratory settings [28]. It provides an accurate automatic sleep staging classification [28] that follows the American Academy of Sleep Medicine guidelines (AASM; [29]). The device has found application in studies involving both clinical and non-clinical populations, allowing also for the auditory closed-loop stimulation of sleep slow oscillations [30,31].

From the DH recordings, we extracted the following classical sleep architecture variables: Sleep Onset Latency in minutes (SOL), Time in Bed (TIB; i.e., total amount of time, in minutes, from lights off to final awakening), Total Sleep Time (TST; i.e., total amount of time, in minutes, from the first appearance of N1 to the final awakening), Actual Sleep Time (AST; i.e., total time spent in sleep states, expressed in minutes), sleep stage proportions over TST (N1%, N2%, N3%, REM%), Wake After Sleep Onset over TST, in minutes (WASO), and Sleep Efficiency (SE%; i.e., percentage of AST over TIB). Moreover, as in [32], we computed an additional set of variables indexing the following features:Sleep continuity: total frequency of awakenings per hour of AST;Sleep stability: arousal frequency per hour of AST (arousals are defined as all transitions to shallower NREM sleep stages and from REM sleep to N1); state transition frequency per hour of TST (state transitions are defined as all transitions from one state to another); frequency of ‘Functional Uncertainty’ periods per hour of TST (‘FU’ periods; defined as periods in which a minimum of three state transitions follow one another with no longer than 1.5 min intervals; [33]);Sleep organization: total time spent in sleep cycles over TST (TCT%), where sleep cycles are defined as sequences of NREM and REM sleep (each lasting at least 10 min) not interrupted by periods of wake longer than 2 min.

### 2.5. Data Analysis

As in previous studies on the topic [20,27], scoring was performed by dividing each text into 30 units. The experimenter assigned one point to each correctly recalled unit (out of 30). Units containing synonyms or with reversal of word order were also considered correct recalls. The immediate test performance at the prose memory task served as a confirmation of participants’ successful memorization of the text, as well as a baseline measure of memory performance. To quantify the degree of memory decay across the retention intervals, we computed the number of text units correctly recalled in the immediate test minus the number of text units correctly recalled in the delayed test (Δunits) as the main dependent variable.

Normality of distribution was checked through the Shapiro–Wilk test. To exclude differences in encoding efficacy between the Sleep and the Wake conditions, a linear mixed model (LMM) was used with Condition (Sleep and Wake) and Text (text 1 and text 2) as factors, and the number of text units correctly recalled in the immediate test as the dependent variable. For delayed performance, we performed a linear mixed model (LMM) with Condition (Sleep and Wake) and Text (text 1 and text 2) as factors, Δunits as the dependent variable, and Sleep Duration (categories: 1 = < 6.5 h, 2 = ≥ 6.5 h) as the covariate. A Pearson’s correlation analysis was also performed to investigate associations between sleep measures and memory performance.

In order to assess between-conditions differences in sleepiness levels (i.e., Karolinska Sleepiness Scale values) in the learning and the re-test phases, we performed a Student’s *t* test for paired samples. The same test was employed to investigate differences in the level of interest, perceived difficulty, and previous knowledge related to the two prose passages.

A priori and a posteriori power analyses were performed through the G*Power software. The analyses were performed using Jamovi (version 2.3.21; The Jamovi Project, 2024); the significance level was set at *p*≤ 0.05.

## 3. Results

### 3.1. Memory Performance

Mixed models of immediate memory performance revealed no significant effect of Condition (F_1,8_ = 0.975, *p* = 0.352) or Text (F_1,8_ = 0.024, *p* = 0.880) and no Condition × Text interaction (F_1,8_ = 0.046, *p* = 0.834), indicating that there were no differences in the level of acquisition between the conditions (Sleep condition: 12.30 ± 4.40; Wake condition: 14.20 ± 4.59) or between the texts (model R^2^ conditional = 0.218).

As for delayed performance (R^2^ conditional = 0.269), an effect of Condition emerged (F_1,15_ = 4.342, *p* = 0.05). Specifically, performance improved across the retention interval spent asleep (Δunits: 1.00 ± 3.40), whereas it deteriorated across the wake interval (Δunits: −1.70 ± 2.21) (Figure 2). No significant effect of Text (F_1,15_ = 0.721 *p* = 0.409) or Sleep Duration (F_1,15_ = 0.199, *p* = 0.662) and no Condition × Text interaction (F_1,15_ = 0.490, *p* = 0.495) emerged.

### 3.2. Sleep Features and Correlations with Memory Performance

Table 1 displays the participants’ sleep measures. Correlation analyses revealed a significant positive correlation between TCT% and Δunits, suggesting that the higher the proportion of time spent in sleep cycles in the sleep episode, the more the participants benefited from sleep in text retrieval (r = 0.749, *p* = 0.032, r^2^ = 0.56). No other significant correlation between sleep variables and memory performance emerged.

### 3.3. Sleepiness Levels

The participants reported higher KSS values in the Sleep than in the Wake condition both in the learning phase (Sleep = 6.00 ± 2.40; Wake = 2.60 ± 1.07; *t* (9) = 5.21, *p* < 0.001; Cohens’ d = 1.646) and in the testing phase (Sleep = 4.50 ± 2.22; Wake = 2.20 ± 1.55; *t* (9) = 2.53, *p*= 0.032; Cohens’ d = 0.802).

### 3.4. Final Questions About the Text

Table 2 displays the participants’ answers to the questions regarding the studied texts. Level of interest and ease of reading were rated as similar for the texts, whereas the participants reported having more prior knowledge about text 1 compared to text 2 (*p* = 0.004). None of the participants had read either text before.

## 4. Discussion

This study aimed to assess the effect of sleep on the consolidation of prose memory, a form of long-term declarative memory very frequently used in everyday contexts, especially in academic and educational settings. Despite its relevance in daily life, the relationship between sleep and prose memory has been surprisingly under-explored. Indeed, although many studies highlighted the positive effect of a retention period spent asleep on performance in simple declarative memory tasks, such as word list learning, it is still unclear whether and to what extent sleep benefits the consolidation of more complex declarative information, such as prose passages.

In this study, we deemed it relevant to include an immediate test in the learning phase for two key reasons. First, it allowed us to rule out the presence of between-conditions differences in the immediate test, ensuring that the performance observed in the delayed test reflected the consolidation processes occurring over the retention interval in S and W, rather than the effectiveness of encoding. Additionally, the immediate test served as a baseline measure enabling us to quantify the possible decline in memory trace during the retention period in both conditions.

The results of this study are consistent with previous findings showing positive effects of sleep on the consolidation of complex declarative memory materials [14,15,20]. In fact, we observed that, in the Sleep condition, the participants recalled more units of text at awakening than they did in the immediate test. Conversely, in the Wake condition, we found a worsening of performance over the retention interval. Therefore, our results suggest that, as far as prose memory is concerned, sleep not only protects memory traces against the decay occurring during periods of wakefulness but also reinforces them. Interestingly, this finding is at variance with data from most studies on word list learning (e.g., [13,34]), in which the sleep effect is generally expressed as a lower decay of memory performance in sleep relative to wake conditions. These different results can be interpreted in an ecological perspective: in fact, as already suggested elsewhere (e.g., [4,35]), the role of sleep in memory consolidation appears to be that of optimizing learning in a manner that is functional to future behavior. Therefore, it can be hypothesized that the sleep-dependent benefits observed in learning are more pronounced in tasks that mimic everyday life activities (such as increasing general knowledge through reading) rather than in classical laboratory tasks that bear scarce resemblance to real-life memory processes. In other words, prose memory tasks would allow for highlighting the ‘reconstructive’ rather than the ‘reproductive’ nature of sleep-related memory processing [4] by triggering, during subsequent sleep, the activation of higher order semantic processing than that engaged after word list learning (e.g., integration of new knowledge into existing networks, extraction of gist).

Furthermore, the correlation analyses detected a significant positive association between the percentage of time spent in sleep cycles (TCT%) and the Δunits. This finding implies that the more organized a sleep episode is, the more effectively it could counteract the natural decay of memory traces, thereby enhancing the ability to recall texts upon awakening. Sleep organization—defined by sleep variables linked to the number and duration of sleep cycles—is known to be crucial for biological and neurophysiological processes [36,37], as well as for long-term memory consolidation [38,39,40]. Indeed, the participants in this study all displayed good sleep quality, as confirmed by the polysomnographic recordings. Moreover, this finding is in line with data from our previous study on prose memory [19], in which we observed that, compared to a condition in which sleep was not preceded by learning, sleep after learning theatrical monologues showed higher continuity, stability, and cyclic organization. Incidentally, we observed a similar pattern of results also with a complex multi-componential task [41]: in this study, sleep cyclic organization was enhanced after pre-sleep training in the Ruzzle game compared to that recorded following a gaming session that minimized the learning demands. Although, in the latter study, the task fell in the implicit memory domain, its analogy to prose memory tasks resides in its complexity and resemblance to real-life learning processes. Taken together, these findings support the idea of a relationship between well-structured sleep and more efficient memory consolidation of complex ecological tasks.

The study results should be considered in the light of some limitations. First, the small sample size and the limited statistical power could affect the generalizability of the findings, and our results require further replication in larger samples. Secondly, the participants reported differing levels of prior knowledge about the two texts used in the experimental conditions. However, we found no effect of the type of text on memory performance in the immediate test. This evidence, together with the fact that the assignment of the texts to the two conditions was balanced between the participants, allows us to rule out an influence of the type of text on memory performance. Furthermore, we found higher levels of sleepiness in the Sleep than in the Wake condition, both in immediate and in delayed testing. This could be attributed to the natural increase in sleep propensity during the evening hours and to possible lingering effects of sleep inertia after awakening [42]. Still, we can rule out an effect of the vigilance levels on either phase of the study. In fact, performance in the immediate test did not differ between the conditions, despite the different vigilance levels. As for delayed recall, the participants performed better after sleep, notwithstanding the higher sleepiness reported in this condition.

In conclusion, this study provides evidence supporting the role of sleep in the consolidation of prose memory, a complex and highly relevant form of declarative memory in daily life. Our findings suggest that sleep not only protects prose memory from decay but also actively enhances it. Future research should build on these results by using larger and different samples, including children, older adults, and individuals with cognitive impairments. Moreover, it would be valuable to investigate how sleep influences not just the quantitative retention of prose memory but also the qualitative reorganization of it through processes of transformation, integration in pre-existing knowledge networks, and extraction of gist [4].

## 5. Conclusions

Unlike studies relying on simple word lists, this research examined how sleep affects our ability to remember more complex declarative information, like prose passages. Our study found that a sleep period helps us to remember prose materials better than an equivalent interval spent awake. Our results confirm that sleep enhances the process of memory retrieval of narrative text, highlighting the importance of sleep for everyday learning and, specifically, for knowledge acquisition.

## Figures and Tables

**Figure 1 brainsci-15-00265-f001:**
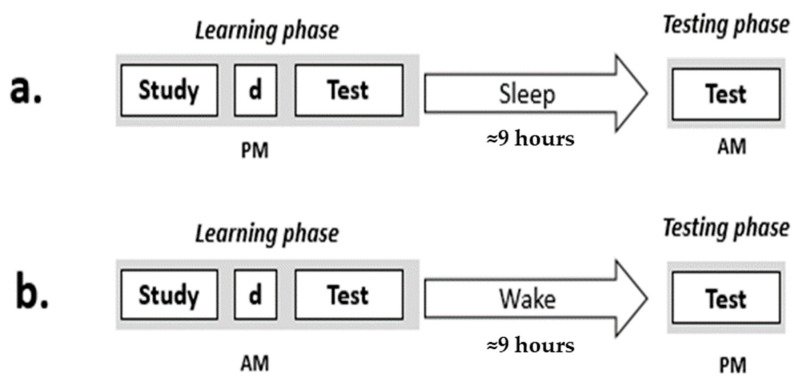
The study procedure. (**a**) Sleep condition procedure, (**b**) Wake condition procedure. ‘d’ refers to a 2 min distractor task. The Sleep condition experiment was conducted at the participant’s home, whereas the Wake condition experiment was performed at the Sleep Laboratory.

**Figure 2 brainsci-15-00265-f002:**
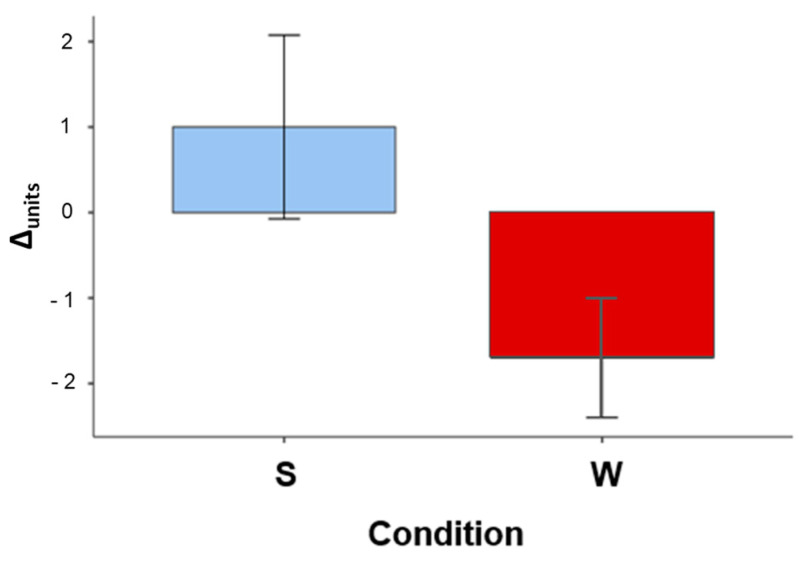
Δunits in the Sleep condition (S) and in the Wake condition (W). Standard deviations are reported.

**Table 1 brainsci-15-00265-t001:** Sleep measures.

Sleep Architecture	
SOL	15.06 ± 4.86
TIB	436.13 ± 58.32
TST	421.69 ± 58.88
AST	404.19 ± 57.40
N1%	5.83 ± 4.75
N2%	45.61 ± 3.96
N3%	22.77 ± 6.04
REM%	24.89 ± 6.40
WASO (min)	16.50 ± 9.38
SEI%	92.59 ± 2.69
Sleep Continuity, Stability, and Organization	
Arousal (fq)	2.921 ± 0.87
Awakenings (fq)	3.123 ± 1.49
State transitions (fq)	15.664 ± 5.00
FU (fq)	1.044 ± 0.58
TCT%	46.30 ± 25.99

Note: Sleep Onset Latency in minutes (SOL), Time in Bed (TIB), Total Sleep Time (TST), Actual Sleep Time (AST), sleep stage proportions over TST (N1%, N2%, N3%, REM%), Wake After Sleep Onset (WASO), Sleep Efficiency (SEI%), Functional Uncertainty periods (FU), total time spent in sleep cycles over TST (TCT%).

**Table 2 brainsci-15-00265-t002:** Participants’ answers to questions regarding the prose passages.

	Text 1	Text 2	* t *	* p *
Q1—Interest	4.00 ± 1.75	3.50 ± 1.00	1.76	0.095
Q2—Perceived difficulty	3.00 ± 2.00	2.00 ± 1.75	1.82	0.085
Q3—Prior knowledge	3.50 ± 2.00	1.00 ± 0.00	3.29	0.004

Notes. Text 1: ‘The Sun’; Text 2: ‘The Sea Otters’. Response alternatives for Q1 ranged from 1 = very interesting to 7 = very boring; for Q2, from 1 = very easy to 7 = very difficult; for Q3, from 1 = very poor knowledge to 7 = very high knowledge.

## Data Availability

Dataset available on request from the authors: the data are not publicly available due to privacy reasons.

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
