# Peer review of "Sleep Benefits Prose Memory Consolidation in University Students"

_brainsci, 2025, doi:10.3390/brainsci15030265_

Round 1
Reviewer 1 Report
Comments and Suggestions for Authors
The authors conducted a very smart, funny and interesting study. More specifically, they benefited from the current literature on the association between sleep and memory consolidation and they build up a new and funny study.
Abstract: please report participants age, age range, and generation ratio, and whether they were psychologically healthy. For the results, please report effect sizes.
Introduction: for the reference is six and seven, please report what kind of pros participants had to memorize and to rehearse after sleep. For knowledge transfer during sleep, the author, the author may give a look at (Brand et al., 2010). Further, for basic information on sleep, the authors may give a closer look at (Sharon et al., 2024).
Please remove the sentence to the best of our knowledge because we suppose that authors report the current and most recent state of the art.
Please put the whole text into past tense because you already completed the study; for instance, writing “we compare memory performance” means that you already did compare; thus, write “we compared memory performance” and so on.
Methods: please report how many people you approached and how you selected these 10 out of how many people you did assess. Please report whether you calculate it a sample size. For the post memory task, how did you make sure that all participants reached the mastery level: or in other words, how did you make sure that all participants did know exactly, completely, and perfectly the prose? Further, is it possible to add the process to the manuscript as supplementary material? For figure one, please add the minutes or hours between the test and re-testing phase. Further for figure one, please clarify that the sleep was assessed at participants home, while in the week condition participants were in the lab.
For the sleep analysis, I verified the publication of Arnold at all; so, it appears that these devices provide reliable data; however, could you please provide some further references?
Results; Please avoid abbreviation such as S or DT or IT because the reviewer or the reader does not memorize all these details. Aas such, please do not overestimate the readers working memory. For table one, please add the meanings of the abbreviations. Further, giving a closer look to the data on table one, one night, suspect that there are large differences in the sleep, duration; perhaps it's worth analyzing the data and splitting participants in short sleepers and long sleepers.
Discussion: this part was nicely crafted. This part was nicely crafted. Please pay attention to the references. All of a sudden there is Conte et al. 2022, and I think it does not fit with the overall requirements for references.
References
Brand, S., Opwis, K., Hatzinger, M., Holsboer-Trachsler, E., 2010. REM sleep is related to the transfer of implicit procedural knowledge following metacognitive learning. Somnologie - Schlafforschung und Schlafmedizin 14(3), 213-220.
Sharon, O., Ben Simon, E., Shah, V.D., Desel, T., Walker, M.P., 2024. The new science of sleep: From cells to large-scale societies. PLoS Biol 22(7), e3002684.
Author Response
The authors conducted a very smart, funny and interesting study. More specifically, they benefited from the current literature on the association between sleep and memory consolidation and they build up a new and funny study.
Abstract: please report participants' age, age range, and generation ratio, and whether they were psychologically healthy. For the results, please report effect sizes.
Participants’ information has been added in the Abstract as well as in the paragraph “Participants” (paragraph 2.1.). Effect sizes have also been added in the Abstract as well as in the Results section.
Introduction: for the reference is six and seven, please report what kind of pros participants had to memorize and to rehearse after sleep. For knowledge transfer during sleep, the author, the author may give a look at (Brand et al., 2010). Further, for basic information on sleep, the authors may give a closer look at (Sharon et al., 2024).
Following the Reviewers’ suggestion, we provided further details about the prose materials adopted in the studies we mentioned. Moreover, we added information about sleep and sleep-dependent knowledge transfer considering the references suggested by the Reviewer.
Please remove the sentence to the best of our knowledge because we suppose that authors report the current and most recent state of the art.
Following the Reviewers’ request, the sentence has been removed.
Please put the whole text into past tense because you already completed the study; for instance, writing “we compare memory performance” means that you already did compare; thus, write “we compared memory performance” and so on.
Done
Methods: please report how many people you approached and how you selected these 10 out of how many people you did assess. Please report whether you calculate it a sample size.
We thank the Reviewer for this suggestion. The paragraph “Participants” (paragraph 2.1.) now includes a detailed description of the recruitment process as well as the results of the a priori and a posteriori power analysis. The limited sample size has also been clearly highlighted at the end of the Discussion section as a limitation to the generalizability of results.
For the post-memory task, how did you make sure that all participants reached the mastery level: or in other words, how did you make sure that all participants did know exactly, completely, and perfectly the prose? Further, is it possible to add the process to the manuscript as supplementary material
For figure one, please add the minutes or hours between the test and re-testing phase Further for figure one, please clarify that the sleep was assessed at participants home, while in the week condition participants were in the lab.
As now specified in the paragraph “Prose memory task” (paragraph 2.3.), participants were asked not to memorize the prose passages word for word, but to report as many text details as possible.
We also specified in the same paragraph that the two texts are currently used to assess language skills in young adult high school students, selected from the reading comprehension section of a test-preparation book for the Test of English as a Foreign Language test manual (TOEFL). As now more clearly explained (same paragraph), it is assumed that the two texts are easily understood and learned by the study sample, composed of students attending higher-level courses of study, such as university. Still, performance at the immediate test served both as baseline measure and as confirmation that the texts had been sufficiently encoded before the retention interval.
As for Figure 1, it has also been revised according to the Reviewer’s suggestion.
For the sleep analysis, I verified the publication of Arnold at all; so, it appears that these devices provide reliable data; however, could you please provide some further references?
Following the Reviewer suggestion, we added references to the studies that employed DREEM in clinical and basic research settings.
Results; Please avoid abbreviation such as S or DT or IT because the reviewer or the reader does not memorize all these details. Aas such, please do not overestimate the readers working memory. For table one, please add the meanings of the abbreviations. Further, giving a closer look to the data on table one, one night, suspect that there are large differences in the sleep, duration; perhaps it's worth analyzing the data and splitting participants in short sleepers and long sleepers.
We recognize that the extensive use of acronyms may impact reading fluency and comprehension. The acronyms have now been replaced and appropriate legends have been added to facilitate the interpretation of the variables in Table 1.
As for the comment on sleep duration, we thank the Reviewer for his observation which prompted us to improve the process of data analysis. The sample has now been split and this variable has been included as covariate in a Linear Mixed Model analysis (which is considered as a more adequate analysis when evaluating covariates within groups, Keselman et al., 2001; Detry & Ma, 2016). Note that there are no substantial differences in the results and that the effect of sleep duration is not significant.
1) Keselman, H.J., Algina, J. and Kowalchuk, R.K. (2001), The analysis of repeated measures designs: A review. British Journal of Mathematical and Statistical Psychology, 54: 1-20. https://doi.org/10.1348/000711001159357
2) Detry, M. A., & Ma, Y. (2016). Analyzing Repeated Measurements Using Mixed Models. JAMA, 315(4), 407–408. https://doi.org/10.1001/jama.2015.19394
Discussion: this part was nicely crafted. This part was nicely crafted. Please pay attention to the references. All of a sudden there is Conte et al. 2022, and I think it does not fit with the overall requirements for references.
We thank the Reviewer for pointing out this typo. We corrected it into the manuscript.
References
Brand, S., Opwis, K., Hatzinger, M., Holsboer-Trachsler, E., 2010. REM sleep is related to the transfer of implicit procedural knowledge following metacognitive learning. Somnologie - Schlafforschung und Schlafmedizin 14(3), 213-220.
Sharon, O., Ben Simon, E., Shah, V.D., Desel, T., Walker, M.P., 2024. The new science of sleep: From cells to large-scale societies. PLoS Biol 22(7), e3002684.

Reviewer 2 Report
Comments and Suggestions for Authors
Thank you for the opportunity to review the manuscript.
The manuscript is fascinating and addresses an important topic. The manuscript is well-written and has many merits. I have some comments that may improve the manuscript.
1) The Introduction section could include more information about decelerative memory, prose memory, and strategies for enhancing prose memory.
2) While the aim is clear, the Authors should add study questions or/ and hypotheses.
3) Lines 70-71: "Participants were selected from a sample of volunteer students recruited via online advertisements." The authors should provide what they mean by online advertisements, which platforms?
4) Did the Authors test the number of correct retained words or just the correct units?
5) Did the Authors test the speed of performance?
6) The levels of prior knowledge should be controlled during the analyses of memory performance.
7) There is no doubt that the sample size was small, which could affect the generalizability of the findings. So the Editor should consider this limitation.
Author Response
Thank you for the opportunity to review the manuscript.
The manuscript is fascinating and addresses an important topic. The manuscript is well-written and has many merits. I have some comments that may improve the manuscript.
1) The Introduction section could include more information about decelerative memory, prose memory, and strategies for enhancing prose memory.
Following the Reviewer’s suggestion, further information on these topics has been added in the Introduction section.
2) While the aim is clear, the Authors should add study questions or/ and hypotheses.
Following the Reviewer’s suggestion, we added the study hypothesis at the end of the introduction section.
3) Lines 70-71: "Participants were selected from a sample of volunteer students recruited via online advertisements." The authors should provide what they mean by online advertisements, which platforms?
As now clarified in the paragraph “Participants” (paragraph 2.1.), the study was presented to the university student population through an online advertisement on major social media platforms (WhatsApp, Instagram).
4) Did the Authors test the number of correct retained words or just the correct units?
Our analysis was based solely on the number of correct units in line with previous studies adopting this kind of procedure (Roediger & Karpicke, 2006; Bäuml et al., 2014). This is now more clearly specified in the manuscript (beginning of paragraph 2.5. “Data analysis”).
5) Did the Authors test the speed of performance?
We chose not to assess performance speed in order to maintain the setting of learning as ecological as possible, e.g., by avoiding external stressors like time constraints. As described in the methods section, participants were given 10 minutes to freely recall the text, allowing participants sufficient time to report as many details as possible.
6) The levels of prior knowledge should be controlled during the analyses of memory performance.
We recognize that the level of prior knowledge differed between the two texts. However, the finding that immediate performance was similar between conditions, together with the fact that the assignation of texts to conditions was balanced between participants, allow us to exclude that prior knowledge had an effect on performance. Furthermore, we actually originally ran a linear mixed model analysis with a stepwise evaluation of predictors and this variable significantly increased both the AIC (Akaike Information Criterion) and BIC (Schwartz’s Bayesian Information Criterion) criteria, which reduces the goodness-of-fit of the model. In light of these considerations, we chose to not include this variable in the analysis. However, this issue is clearly highlighted as a limitation at the end of the Discussion section.
7) There is no doubt that the sample size was small, which could affect the generalizability of the findings. So the Editor should consider this limitation.
We agree with the reviewer that the small sample size is a limitation of this study. A priori and a posteriori power analysis are now reported in the manuscript (paragraph 2.1. “Participants”) and this limitation is clearly highlighted at the end of the Discussion section.

Round 2
Reviewer 2 Report
Comments and Suggestions for Authors
Well-done. The authors have addressed all my comments.